# Superconducting spintronic tunnel diode

E. Strambini [1✉], M. Spies[1✉], N. Ligato [1], S. Ilić[2], M. Rouco[2], Carmen González-Orellana[2], Maxim Ilyn[2], Celia Rogero[2,3], F. S. Bergeret[2,3], J. S. Moodera[4], P. Virtanen[5], T. T. Heikkilä [5] & F. Giazotto[1✉]

Diodes are key elements for electronics, optics, and detection. Their evolution towards low dissipation electronics has seen the hybridization with superconductors and the realization of supercurrent diodes with zero resistance in only one direction. Here, we present the quasi-particle counterpart, a superconducting tunnel diode with zero conductance in only one direction. The direction-selective propagation of the charge has been obtained through the broken electron-hole symmetry induced by the spin selection of the ferromagnetic tunnel barrier: a EuS thin film separating a superconducting Al and a normal metal Cu layer. The Cu/EuS/Al tunnel junction achieves a large rectification (up to ~40%) already for a small voltage bias (~200 μV) thanks to the small energy scale of the system: the Al superconducting gap. With the help of an analytical theoretical model we can link the maximum rectification to the spin polarization ($P$) of the barrier and describe the quasi-ideal Shockley-diode behavior of the junction. This cryogenic spintronic rectifier is promising for the application in highly-sensitive radiation detection for which two different configurations are evaluated. In addition, the superconducting diode may pave the way for future low-dissipation and fast super-conducting electronics.

[1] NEST, Istituto Nanoscienze-CNR and Scuola Normale Superiore, I-56127 Pisa, Italy. [2] Centro de Física de Materiales (CFM-MPC) Centro Mixto CSIC-UPV/EHU, E-20018 Donostia-San Sebastián, Spain. [3] Donostia International Physics Center (DIPC), E-20018 Donostia–San Sebastián, Spain. [4] Physics Department and Plasma Science and Fusion Center, Massachusetts Institute of Technology, Cambridge, MA 02139, USA. [5] Department of Physics and Nanoscience Center, University of Jyväskylä, P.O. Box 35 (YFL), FI-40014 Jyväskylä, Finland. ✉email: elia.strambini@sns.it; maria.spies@nano.cnr.it; francesco.giazotto@sns.it

D iodes are non-linear and non-reciprocal circuits in which a lack of spatial inversion symmetry provides a strongly direction-selective electron transport. In the long and successful history of diodes, the material search for this symmetry breaking has been mainly focused on semiconducting and metallic junctions. However, owing to their large energy gap, semiconductors cease to work at the sub-Kelvin temperatures relevant for emerging cryogenic electronics[1] and ultrasensitive detection, especially at sub-THz frequencies[2]. This problem could be partially solved by using low-dimensional structures like quantum dots, which do exhibit current rectification[3,4]. Given that the electron-hole symmetry in quantum dots can be tuned only within the level of a single quantum channel, the impedance of such systems tends to be high, and the rectified currents thereby very low, limiting the value of this approach. Superconductors would be ideal candidates for the realization of cryogenic diodes and detectors due to their intrinsically low impedance, and the lower energy scales of the superconducting gap ($\sim$meV) compared to semiconductors ($\sim$eV). Still, the implementation of a superconducting diode turns out to be difficult since it requires breaking of the electron-hole symmetry, whereas the BCS superconducting state is, by definition, electron-hole symmetric. Recently, supercurrent diodes have been engineered with metallic superlattices in strong magnetic fields, offering the required lack of spatial inversion[5,6], with thin films patterned with nanoholes[7], with unconventional Josephson junctions[8–11] and with superconductors with large spin-orbit interactions[12–14]. Alternative approaches of realizing a superconducting diode are also possible with the quasi-particle counterpart in spin-selective tunnel junctions. When both spin filtering and splitting are present it is possible to break the electron-hole symmetry of the system and generate direction-selective electron transport[15]. Ferromagnetic insulators (FI) like Eu-based chalcogenides combined with superconductors (S) offer bright perspectives for the realization of this family of superconducting spintronic technologies[16]. Devices based on thin films of FI/S bilayers showing ideal spin filtering and spin splitting[17,18] have been already demonstrated in a number of seminal experiments performed on EuS/Al-based tunnel junctions[17,19,20]. Here, we show a superconducting diode based on a spin-selective Al/EuS/Cu tunnel junction. The observed direction-selective electron transport is at the basis of charge rectification and thermoelectricity[21–23] and makes the design of the present superconducting spintronic device a promising approach for the implementation of biasless ultrasensitive THz detectors[24].

The working principle and device characteristics of the normal metal-ferromagnetic insulator-superconductor (N/FI/S) tunnel junction, central to this paper, are shown in Fig. 1. The schematic of the device structure and measurement configurations for the tunnel spectroscopy can be seen in panel (a). A N strip of Cu and a S strip of Al are oriented perpendicular to one another forming a cross-bar, and are separated by a FI barrier of EuS (see Methods for fabrication details). The EuS layer induces a spin splitting by an energy with magnitude ($h$) in S through interface exchange interaction[19,25,26], and its FI nature causes a spin filtering ($P$) of the electron tunneling across the junction. The former results in an opposite energy shift for the BCS density of states (DoS) of the two spin species, as sketched in Fig. 1(c), while the latter forms a tunneling barrier with different heights for the two spin species. This twofold effect can be probed experimentally by measuring the differential conductance of the tunnel junction and leads to qualitative changes in the system's transport characteristics[19,27,28]. An example of a tunneling conductance measurement as a function of bias voltage across the N/FI/S junction is shown in Fig. 1(d). At small voltages ($|V| \lesssim 200\,\mu$V) the conductance is strongly suppressed due to the lack of states within the superconducting energy

gap. At higher bias voltages, four distinct peaks can be observed in total, corresponding to the four peaks of the two BCS DoS at $e|V| = \Delta \pm h$. The different amplitudes of the conductance peaks stem from the spin filtering $P$, promoting one spin channel with respect to the other. All these parameters can be extracted by fitting the conductance with a numerical model (see Eq. (3) and (14) in the "Methods" for details on the model) that takes into account the spin splitting, spin relaxation, and orbital depairing due to the magnetic field[28], as shown by the red curve in Fig. 1(d). Additionally, the application of an external magnetic field can strengthen the polarization of the EuS layer and enhance both $h$ and $P$, as shown in Fig. 1(e) and (f). Notably, thanks to the ferromagnetism of the EuS, both a sizable splitting and polarization are observed even at zero field ($h_0 \simeq 0.025\Delta$, $P_0 \simeq 0.2$, and $\Delta = 370$ $\mu$eV). These vanish at the EuS coercive field ($\simeq$10 mT).

## Results and discussion

**Measurement configurations**. In the following, two measurement configurations (sketch in Fig. 1a) have been adopted to quantify the diode characteristics. In configuration (i) the current flows from the S to the N layer, thereby effectively crossing the junction. A conventional rectification is observed in this case. In configuration (ii) the current flows along the N strip, and a transverse rectification is observed. In both cases the voltage drop is measured from the S to the N layer across the tunnel junction. Notably, in configuration (ii) the current and voltage paths are decoupled, providing more flexibility in projecting the impedance of the device, with advantages for the integration of an antenna and the implementation of a detector. Measurements of the two configurations are compared and discussed. A typical current vs. voltage ($I(V)$) characteristic of the tunnel junction shows a conventional rectification, as can be seen in Fig. 2. It corresponds to measurement configuration (i) in voltage bias. The current bias configuration is considered in the Supplementary Information (section I) together with an alternative choice of material layers, namely EuS/Al/AlO$_x$/Co (section II).

**Rectification**. The presence of the superconducting gap can be clearly recognized in the $I(V)$ characteristic displayed in Fig. 2(b): the absence of current flow at low bias, and an Ohmic behavior for relatively large voltage ($eV \gg \Delta$). In an intermediate voltage range, non-linearities and non-reciprocity appear, which can be visualized in the symmetric and antisymmetric parts of the $I(V)$ characteristic. They are defined as $I_{\mathrm{Sym}} = \frac{I(V)+I(-V)}{2}$ and $I_{\mathrm{Antisym}} = \frac{I(V)-I(-V)}{2}$, and are shown in Fig. 2(c). The sizable $I_{\mathrm{Sym}}(V)$ already suggests an efficient charge rectification i.e. the capability to convert an AC input into a DC output signal. Rectification ($R$) of a circuit can be defined as the ratio between the difference of the forward and backward current divided by the sum of the two, $R(V) = \frac{I(V)+I(-V)}{I(V)-I(-V)} = I_{\mathrm{Sym}}/I_{\mathrm{Antisym}}$, and is shown in Fig. 2(d). For ideal rectifiers $R = 1$, while for $R = 0$ no rectification is present. In the junction a $R$ up to $\sim 40\%$ can be achieved in the intermediate voltage range ($eV \sim \Delta$). This upper limit, equivalent to the polarization $P$ of the EuS junction can be understood using a simple analytical model for the N/FI/S tunnel junctions, which neglects spin-dependent scattering and orbital depairing. Within these approximations the $I(V)$ tunneling current can be simplified to the instructive expression:

$$I(V) = I_S\left(e^{eV/(k_B T)} - 1\right) + I_S\left[\cosh\left(\frac{eV}{k_B T}\right) - 1\right](P - 1). \quad (1)$$

The current scale $I_S$ depends on the physical characteristics of the device, such as the normal-state resistance, superconducting energy gap and the exchange field, as detailed in the Methods

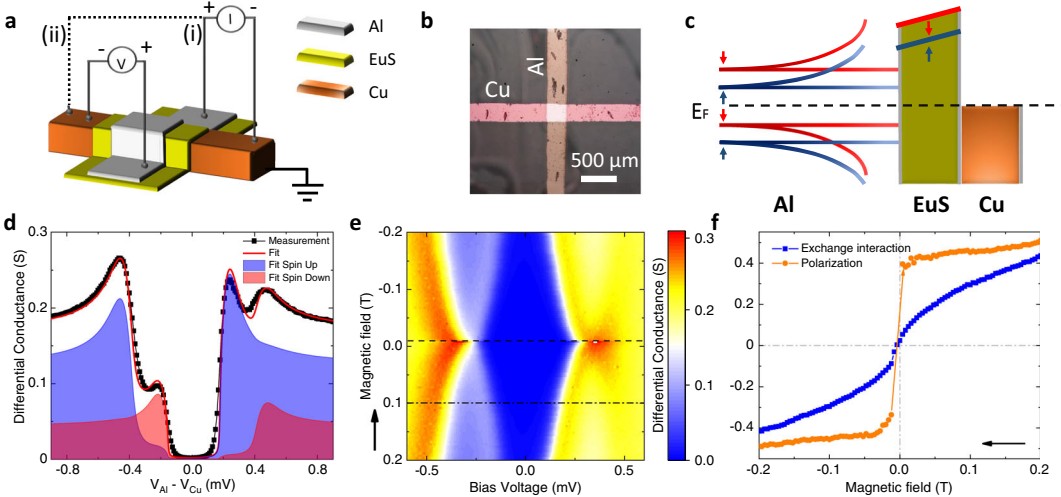

**Fig. 1 Working principle and characteristics of the superconducting tunnel diode. a** Schematic of the device structure: a Cu strip (orange) is covered by a EuS layer (green) and a perpendicular Al strip (gray). Measurement setups: The electric current is applied (i) from the Al to the Cu strip or (ii) along the Cu strip. The voltage drop is measured between the Al and the Cu strip on the remaining two wires of the four-wire set-up. **b** Visible light microscopy image of the device. **c** Schematic of the DoS along the vertical axis of the structure (Al/EuS/Cu from top to bottom). The dashed line indicates the Fermi level. Note that the EuS layer induces spin splitting in the superconducting DoS, and spin filtering thanks to the different heights of the tunnel barrier for the two spin species. The red (blue) line corresponds to the spin up (down) DoS in the Al layer. **d** Exemplary differential conductance (black) measured as a function of voltage across the junction at an applied external magnetic field $B$ of 0.1 T at $\simeq 100$ mK. By employing a numerical model (detailed in the Methods section, Eqs. (3) and (14)), the fit for the differential conductance (red) and the contributions of the spin up (light blue) and spin down (light red) electrons were calculated with these fitting parameters: $\Delta_0 = 0.33$ meV, $h = 0.32\Delta_0$, $P = 0.48$, $\Gamma = 0.01\Delta_0$, $T = 300$ mK. **e** Color map of the differential conductance $dI/dV(V)$ measured for $B$ ranging from $-0.2$ T to 0.2 T. The sweep direction is indicated by the arrow. The data in panel **d** corresponds to the dash-dotted line ($B = 0.1$ T). The coercive field at the temperature of this measurement (100 mK) corresponds to $-9$ mT, indicated by a dashed line. **f** Exchange field ($h$) induced in the superconducting Al strip (blue) and polarization ($P$) of the EuS tunnel barrier as a function of the external magnetic field $B$. Both quantities are extracted from the best fitting results of the data as shown in panel **d**. The sweep direction is again indicated by an arrow.

section, Eq. (4). The expression is valid at low temperatures ($k_B T \ll h$) and voltages ($e|V| \ll \Delta - h$). Note that subgap states due to inelastic scattering can provide an additional contribution to the current $\delta I$, which also satisfies $\delta I(V) \neq -\delta I(-V)$, and becomes particularly important at very low temperatures (see Methods for more details). Equation (1) is composed by two elements. The first one represents the Shockley ideal diode equation[29] and dominates when $P$ is close to unity. It describes the asymmetric I(V) curves characteristic of diodes. The second contribution is the first correction to an ideal diode due to the non-ideal spin polarization. This yields the simple result for the rectification, $R = P \tanh[eV/(2k_B T)]$. The maximum rectification at $|eV| \gtrsim 2k_B T$ is hence dictated by the spin filtering efficiency $P$. Due to the strong asymmetry induced by the spin filtering for this specific junction, $R$ is maximized around 225–280 $\mu$V where it obtains values as high as $\sim$40%, in good agreement with the polarization value extracted from the $dI/dV$ fits (see Fig. 1(f)).

**Transverse rectification.** Notably, the geometry of the device together with the small resistance of the tunnel junction allows for the implementation of a "three-terminal" diode in which the path of the rectified signal (in this case the voltage) is decoupled with respect to the excitation current ($I_H$) path. This corresponds to measurement configuration (ii) and is sketched in Fig. 3(a) and (b). The device is operated with a current bias $I_H$ applied along the Cu bottom lead, while the voltage drop is measured orthogonal to it. At the junction, $I_H$ can partially flow in the S lead and generate a voltage due to the non-symmetric response of the junction to the flowing current (see the sketch displayed in Fig. 3(a)). The resulting measured voltage $V_{sym}$, symmetrized to discard the trivial Ohmic component originated in the Cu lead, is shown in Fig. 3(c) for different magnetic fields. A monotonic increase of $V_{sym}(I_H)$ is visible and more pronounced at large fields

due to the larger $h$ and $P$ of the junction. Notably, a sizable transverse rectification is present also at zero field thanks to the ferromagnetic nature of the FI layer. This characteristic is especially relevant for applications since no additional lines to generate an external magnetic field need to be integrated while operating the device. On the other hand, at the EuS coercive field ($\simeq 14$ mT at base temperature) the rectified signal is not visible, confirming the spintronic nature of this effect.

From the experimental I(V) characteristics of the diode shown in Fig. 2(b) it is possible to model the transverse response of the diode (see Methods for calculation details). Our theoretical calculations compared with the data shown in the inset of Fig. 3(c) are in agreement with the experiment but are generally larger than the measured data by about $\sim$ 30%. This difference likely stems from the thermoelectric effect that, similar to rectification, is also present in the junction with lack of electron-hole symmetry[21].

**Thermodynamic considerations.** From a thermal model that considers the Joule heating induced by $I_H$, we can estimate the resulting thermovoltage and find that it is smaller and of the opposite sign with respect to the rectification voltage (see section IV of the Supplementary Information for more details), therefore confirming the presence of thermoelectricity in the junction. Notably, the relative amplitude of the two effects depends on the length of the tunnel junction, with transverse rectification dominating for junctions longer than $\sim$100 $\mu$m (this length scale depends on various sample specific parameters as described by Eq. (S18) in the Supplementary Information). A similar analysis can be carried out on the rectified current of configuration (i). Also in this case, the excitation voltage can overheat the S metal and generate a thermoelectric current opposite to the rectified signal.

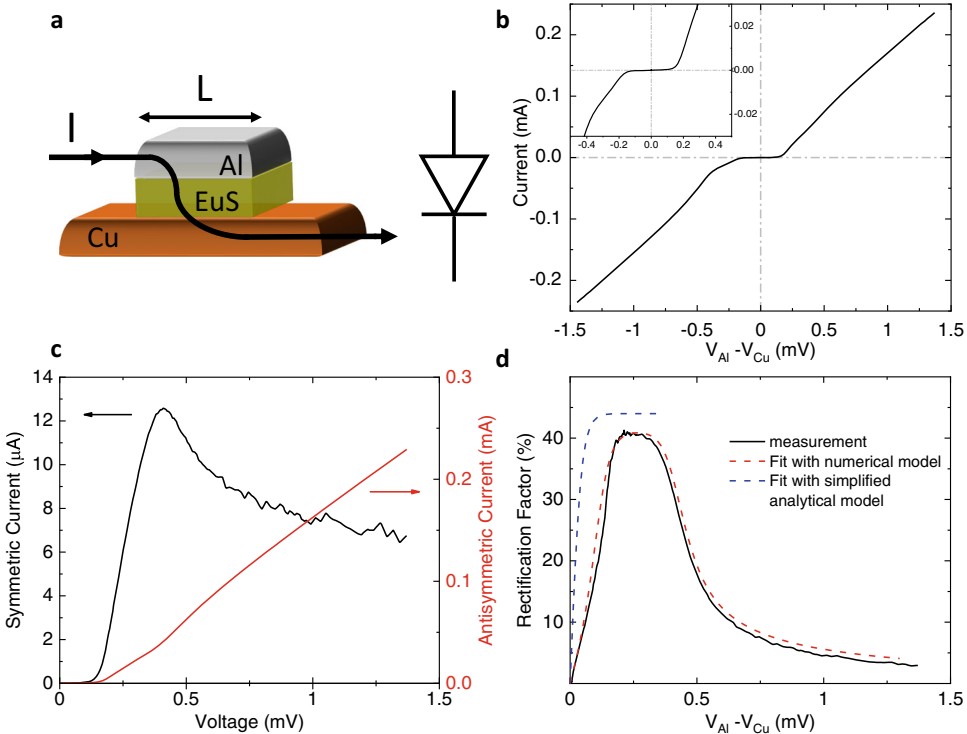

**Fig. 2 Rectification of the superconducting tunnel diode. a** Schematic of the N/FI/S tunnel junction. The path of the tunneling current is indicated by the black line and its arrows. In terms of electronic circuit elements this junction behaves like the indicated diode: the current flows preferentially from the Al layer to the Cu layer while the reverse flow is inhibited. **b** Current-to-Voltage ($I(V)$) characteristics of the junction measured at $T \simeq 50$ mK, $B = 0.1$ T in the four-wire configuration (i). **c** Symmetric and antisymmetric parts of the $I(V)$ characteristic of panel **c** showing a sizable symmetric component of the current. **d** Rectification coefficient $R(V) = I_{Sym}(V)/I_{Antisym}(V)$ evaluated from **e** (black line) along with the comparison with the rectification extracted from the approximated analytical model $R = P\tanh[eV/(2K_BT)]$ (blue line) and the full numerical ones (red line). Details of the numerical model can be found in the Methods section, specifically in Eqs. (3) and (14). Notice the good qualitative agreement with the simplified model predicting the saturation at $R \simeq P \sim 40\%$ at 225–280 $\mu$V. The model ceases to work when $eV \gtrsim \Delta - h \sim 250\,\mu$eV. The discrepancy between the analytical model and the experiment mostly comes from weak inelastic scattering, and to a lesser extent from spin relaxation and orbital depairing.

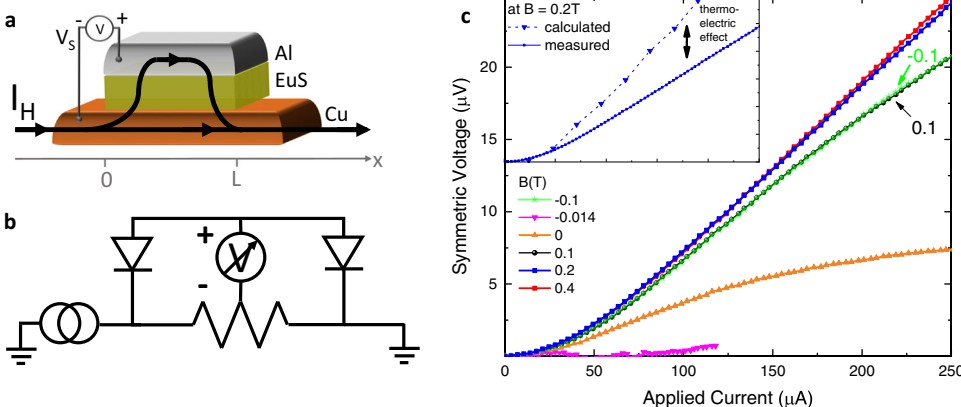

**Fig. 3 Transverse rectification of the superconducting tunnel diode. a** Schematic of the N/FI/S tunnel junction and current path. A biasing current $I_H$ is applied from one end of the Cu strip to the other, while the voltage drop across the junction is measured from the Al contact to the Cu one. The path of tunneling current is indicated by the black line and its arrows. **b** Electronic circuit diagram of the setup. Note that the EuS layer effectively acts as a twofold rectifier for the distributed incoming and outgoing currents tunneling through the FI barrier. **c** Transverse voltage drop $V_{sym}(I_{Cu})$ measured across the barrier as a function of the applied current $I_H$ at different $B$ and at 50 mK. Note that even at zero applied magnetic field (orange curve) a voltage drop occurs, while at the coercive field ($B \simeq -14$ mT) the signal is zero due to the non-polarization of the EuS layer. The $V(I)$ was symmetrized in order to discard the Ohmic (linear) component originating from the N lead. In the inset, the $V_{sym}$ measured at 0.2 T is compared with the calculated data points obtained through a theoretical model of the circuit and using the rectification value obtained from the experimental data.

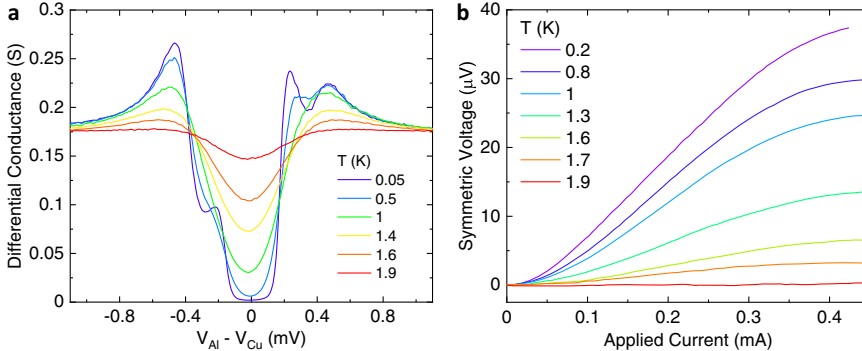

**Fig. 4 Temperature dependence of the superconducting rectifier. a** Differential conductance vs. voltage of the junction measured for different temperatures from 50 mK to 1.9 K. **b** Temperature evolution of the transverse rectification voltage vs. biasing current. Both measurements are performed at $B = 0.1$ T.

Figure 4 shows the temperature dependence of the differential conductance and the transverse rectification voltage of the discussed tunnel junction. Notably, despite the evident thermal broadening of the $dI/dV(V)$ (see Fig. 4(a)), the transverse rectification is only marginally affected below 1 K (see Fig. 4(b)), making the effect very robust even at a temperature up to nearly half of the Al critical temperature ($T_C \simeq 2.3$ K). However, for temperatures larger than $T_C/2$, a clear damping of the signal is visible with measurable effects up to ~1.9 K. This high temperature range of operation makes our superconducting tunnel diode appealing for superconducting electronics schemes where robustness against temperature fluctuations is desirable. Moreover, this behavior is expected to hold for other superconducting materials. There are several FI/S bilayer systems whose $T_C/2$ lies above 4 K (for instance, GdN/NbN bilayers[30]). These materials have the advantage that they can be operated at standard $^4$He cryogenic temperatures and deposited with large-scale sputtering systems.

In conclusion, we have shown the capabilities of a N/FI/S tunnel junction to function both as a conventional diode (i) and as a transverse rectifier (ii). The transverse rectifier benefits from a lower impedance and a direct decoupling between the AC excitation line (the antenna) and the DC sensing line. This advantage allows for more flexibility in the design of the device impedance when compared to configuration (i) for optimizing the impedance matching between the rectifier and the photon absorber, towards optimal quantum efficiency. Both superconducting rectifiers can be operated in zero applied magnetic field showing similar detection sensitivities and noise equivalent powers estimated to be $\sim 1 \times 10^{-12}\mathrm{W}/\sqrt{\mathrm{Hz}}$ with a room temperature voltage amplifier. On the other hand, promising sensitivities up to ~$2 \times 10^3$ A/W and noise equivalent powers down to $\sim 1 \times 10^{-19}\mathrm{W}/\sqrt{\mathrm{Hz}}$ have been estimated for configuration (i) with room-temperature current amplifiers (see section III of the Supplementary Information for details on the analysis). Such very low NEP can be limited by the thermal Johnson–Nyquist noise with an upper bound of $\sim 1 \times 10^{-16}\mathrm{W}/\sqrt{\mathrm{Hz}}$ estimated at 100 mK. This is a step towards the development of detectors in the THz region contributing to the terahertz gap closure.

Besides detection and rectification, this device can be used also for other conventional diode functionalities, but at much lower voltage and thereby much lower dissipation levels than conventional semiconductor-based diodes. Such applications include mixers, reverse current regulators, voltage clamping, and more passive superconducting electronics[1]. Further functionalities can also be expected with more complicated structures containing several EuS or Al layers[31].

## Methods

**Sample fabrication and transport measurements.** The samples are cross-bars made by electron-beam evaporation employing an in-situ shadow mask. The structures consist of a glass substrate on which the layers of Cu(20)/ EuS(2)/ Al(4)/ Al$_2$O$_3$(13) are deposited sequentially (thicknesses in nm). The overlap between the Al and the Cu strip has an area of $300 \times 300\ \mu m^2$. The tunneling spectroscopy is carried out at cryogenic temperatures down to 50 mK in a filtered cryogen-free dilution refrigerator. The $I(V)$ characteristics are obtained from DC four-wire measurements, as sketched in Fig. 1(a), and are used to calculate the differential conductance via numerical differentiation.

**Diode equation.** The $I(V)$ characteristic of the spin-polarized junction to a spin-split superconductor is given by (here, $e = k_B = \hbar = 1$ for brevity)

$$I(V) = \sum_\sigma G_\sigma \int d\epsilon N_\sigma(\epsilon)[f_0(\epsilon - V) - f_0(\epsilon)], \qquad (2)$$

where $\sigma = \pm 1$ for spin up/down, $G_\sigma = G_0(1 + \sigma P)$ is the spin-dependent tunneling conductance, $N_\sigma = (N_0 + \sigma N_z)/2$ is the spin-dependent density of states, $f_0(\epsilon) = [\exp(\epsilon/T) + 1]^{-1}$ is the Fermi function, $G_0$ is the normal-state tunneling conductance, $N_{0/z}$ is the spin average/difference density of states, and $P \in [-1, 1]$ is the spin polarization. Carrying out the sum over the spin yields

$$I(V) = G_0 \int d\epsilon [N_0 + PN_z][f_0(\epsilon - V) - f_0(\epsilon)]. \qquad (3)$$

The distribution function factor can be simplified as

$$f(\epsilon - V) - f(\epsilon) = \frac{1}{e^{(\epsilon-V)/T} + 1} - \frac{1}{e^{\epsilon/T} + 1} = \frac{1 - e^{-V/T}}{1 + e^{-V/T} + e^{-\epsilon/T} + e^{(\epsilon-V)/T}}.$$

Because of the gap in the $N_0$ and $N_z$ functions, this needs to be evaluated only for $\epsilon > \Delta - h$ and for $\epsilon < -\Delta + h$. If $V \ll \Delta - h$, for the positive energies the last term in the denominator is larger than the others so we may approximate

$$f(\epsilon - V) - f(\epsilon) \approx (1 - e^{-V/T})e^{-(\epsilon-V)/T}.$$

On the other hand, for negative energies the third term in the denominator is larger than the others and we may approximate

$$f(\epsilon - V) - f(\epsilon) \approx (1 - e^{-V/T})e^{-\epsilon/T}.$$

In the absence of spin relaxation or orbital depairing, the spin-dependent DoS is

$$N_0 + PN_z = \mathrm{Re}\left[ \frac{|\epsilon + h|}{\sqrt{(\epsilon + h)^2 - \Delta^2}} \frac{1 + P}{2} + \frac{|\epsilon - h|}{\sqrt{(\epsilon - h)^2 - \Delta^2}} \frac{1 - P}{2} \right],$$

we get the current to the form

$$I = \frac{G_0}{2}(1 - e^{-V/T}) \left[ \int_{\Delta - h}^\infty \frac{(\epsilon + h)(1 + P)}{\sqrt{(\epsilon + h)^2 - \Delta^2}} e^{-(\epsilon - V)/T} d\epsilon + \int_{\Delta + h}^\infty \frac{(\epsilon - h)(1 - P)}{\sqrt{(\epsilon - h)^2 - \Delta^2}} e^{-(\epsilon - V)/T} d\epsilon \right.$$
$$\left. - \int_{-\infty}^{-\Delta - h} \frac{(\epsilon + h)(1 + P)}{\sqrt{(\epsilon + h)^2 - \Delta^2}} e^{\epsilon/T} d\epsilon - \int_{-\infty}^{-\Delta + h} \frac{(\epsilon - h)(1 - P)}{\sqrt{(\epsilon - h)^2 - \Delta^2}} e^{\epsilon/T} d\epsilon \right]$$

Shifting the energies by the spin-splitting field up and down, and reverting the sign of the energy in the latter two integrals yields

$$I = \frac{G_0(1 - e^{-V/T})}{2} [(1 + P)e^{(h+V)/T} + (1 - P)e^{(V-h)/T} + (1 + P)e^{-h/T} + (1 - P)e^{h/T}]$$
$$\times \underbrace{\int_\Delta^\infty \frac{\epsilon e^{-\epsilon/T}}{\sqrt{\epsilon^2 - \Delta^2}} d\epsilon}_{=\Delta K_1(\Delta/T)},$$

where $K_1(\Delta/T) \approx \sqrt{\pi/2} e^{-\Delta/T} \sqrt{\frac{T}{\Delta}}$ is the Bessel $K$ function and the latter

approximation is valid for $\Delta \gg T$. Let us define

$$I_S \equiv G_0 \Delta K_1\left(\frac{\Delta}{T}\right) e^{h/T}. \tag{4}$$

Now rearranging terms in the expression for the current allows us to write it as

$$I(V) = I_S(e^{V/T} - 1) + I_S e^{-2h/T}(1 - e^{-V/T})$$
$$+ I_S(1 - e^{-2h/T})\left[\cosh\left(\frac{V}{T}\right) - 1\right](P - 1). \tag{5}$$

This also yields the rectification

$$R = P \tanh\left(\frac{h}{T}\right) \tanh\left(\frac{V}{2T}\right). \tag{6}$$

For $h \gg T$ we get Eq. (1) and the corresponding simplified expression for $R$ quoted in the main text.

**Corrections to the current due to subgap states.** Inelastic scattering introduces subgap states, which can be well described within the Dynes model[32]. At low energies ($\epsilon < \Delta - h$), a weak Dynes parameter $\Gamma \ll \Delta - h$ introduces a correction to the superconducting density of states given as $\delta N_\sigma(\epsilon) = \frac{\Gamma}{\rho_\sigma}(1 + \frac{\epsilon_\sigma^2}{\rho_\sigma^2})$. Here $\epsilon_\sigma = \epsilon + \sigma h$ and $\rho_\sigma = \sqrt{\Delta^2 - \epsilon_\sigma^2}$. Combining this with Eq. (2), we find the following correction to the current, valid at low temperatures and for voltages $V < \Delta - h$:

$$\delta I = \Gamma G_0[F_{asym}(eV, h) + P F_{sym}(eV, h)]. \tag{7}$$

Here we introduced the functions $F_{asym}(eV, h) = \frac{1}{2}[F(eV + h) + F(eV - h)]$, $F_{sym}(eV, h) = \frac{1}{2}[F(eV + h) - F(eV - h) - 2F(eV)]$, with $F(x) = x/\sqrt{\Delta^2 - x^2}$. For small voltages and weak exchange field, $h, eV \ll \Delta$, we may approximate $\delta I \approx \frac{\Gamma eV}{\Delta}[1 + \frac{3}{2}\frac{eVh}{\Delta^2}]$.

Taking into account the correction $\delta I$ together with Eq. (1), the expression for the rectification coefficient $R$ becomes ($h \gg k_B T$)

$$R = P \frac{2\sinh^2 \frac{eV}{2k_B T} + \xi F_{sym}(eV, h)}{\sinh \frac{eV}{k_B T} + \xi F_{asym}(eV, h)}, \tag{8}$$

where $\xi = \frac{G_0 \Gamma}{I_S} \sim \frac{\Gamma}{\Delta} e^{\Delta/(k_B T)}$. If the temperature is high-enough, $k_B T \gg \Delta/\log(\frac{\Delta}{\Gamma})$, we have $\xi \ll 1$, and inelastic scattering can be neglected. In this case we obtain the expression shown in the main text: $R = P \tanh[eV/(2k_B T)]$. However, in the opposite regime of very low temperatures, $k_B T \ll \Delta/\log(\frac{\Delta}{\Gamma})$, we find that $\delta I$ actually provides the dominant contribution to the current. In that case, $R = P F_{sym}/F_{asym}$. Note that in both regimes the maximal rectification coefficient is given by $R_{max} = P$. In the first regime, the maximum is reached at $eV \sim k_B T$, whereas in the second it is at $eV \sim \Delta - h$.

**Model for the density of states (DoS).** In the calculation of the $I(V)$ characteristics the density of states of the superconductor, $\mathcal{N}_\sigma(\epsilon)$, plays a central role. We obtain it from the quasiclassical Green's functions (GFs), $\check{g}$, defined in the Nambu $\otimes$ spin space. These are $4 \times 4$ matrices that satisfy the normalization condition, $\check{g}^2 = 1$. Here the "check" symbol, $\check{\cdot}$, indicates $4 \times 4$ matrices.

In the bulk of a dirty superconductor with a constant exchange field aligned along a given axis, the *retarded* quasiclassical GFs fulfill the following Usadel equation[28,31,33]:

$$[i(\epsilon + i\Gamma)\hat{\tau}_3 + ih\hat{\tau}_3\hat{\sigma}_z - \Delta\hat{\tau}_1 - \check{\Sigma}, \check{g}] = 0. \tag{9}$$

Here, $\epsilon$ is the energy, $\Gamma$ is a small energy term known as the Dynes parameter[32], $h$ stands for the strength of the exchange field, $\Delta$ is the self-consistent superconducting order parameter and $\hat{\tau}_i$ and $\hat{\sigma}_a$ label the Pauli matrices spanning Nambu and spin space, respectively. Direct product between Pauli matrices spanning different spaces is implied, and identity matrices, $\hat{\tau}_0$ and $\hat{\sigma}_0$, are obviated. The square brackets, $[\cdot, \cdot]$, stand for commutation operation and $2 \times 2$ matrices are indicated with a $\hat{\cdot}$ symbol. A typical value of the Dynes parameter is $\Gamma \sim 10^{-3}\Delta$ and its importance is twofold: first it avoids analytical problems in the computation of the GFs and second it models the effect of non-elastic processes in the superconductor. The $\check{\Sigma}$ matrix is the self-energy that consists of three contributions:

$$\check{\Sigma} = \check{\Sigma}_{so} + \check{\Sigma}_{sf} + \check{\Sigma}_{orb}. \tag{10}$$

From left to right, these are the spin relaxation due to spin-orbit coupling, the spin relaxation due to spin-flip events and the orbital depairing due to external magnetic fields, respectively. Explicitly, each contribution within the relaxation time approximation, reads:

$$\check{\Sigma}_{so} = \frac{\hat{\sigma}_a \check{g} \hat{\sigma}_a}{8\tau_{so}}, \quad \check{\Sigma}_{sf} = \frac{\hat{\sigma}_a \hat{\tau}_3 \check{g} \hat{\tau}_3 \hat{\sigma}_a}{8\tau_{sf}}, \quad \check{\Sigma}_{orb} = \frac{\hat{\tau}_3 \check{g} \hat{\tau}_3}{\tau_{orb}}. \tag{11}$$

Here $\tau_{so}$, $\tau_{sf}$ and $\tau_{orb}$ stand for spin-orbit, spin-flip and orbital depairing relaxation times, respectively, and we sum over repeated indices. We estimate the orbital

depairing in the superconducting layer due to an applied magnetic field as[34,35]:

$$\tau_{orb}^{-1} \equiv \left(\frac{\pi d \xi_0 B}{\sqrt{6}\Phi_0}\right)^2 \Delta_0, \tag{12}$$

where $\Phi_0$ is the quantum of magnetic flux, $d$ stands for the width of the superconducting layer, $B$ is the applied magnetic field, $\Delta_0$ is the superconducting gap at zero field ($T = 0$ and $h = 0$) and $\xi_0$ is the superconducting coherence length.

In addition to Eq. (9), the value of the superconducting gap is related to the quasiclassical GFs via the self-consistent equation,

$$\Delta = \frac{\lambda}{8i} \int_{-\Omega_D}^{\Omega_D} d\epsilon \, \text{Tr}\left[\hat{\tau}_1 \check{g}\right], \tag{13}$$

where the trace runs over the Nambu $\otimes$ spin space, $\lambda$ is the coupling constant and $\Omega_D$ is the Debye cutoff energy.

From Eqs. (9), (13) and the normalization condition we compute the value of $\check{g}$, from which the the spin average/difference density of states, $\mathcal{N}_{0/z}$, can be directly calculated:

$$\mathcal{N}_{0/z}(\epsilon) = \frac{1}{2} \text{Re}\left[\text{Tr}\left(\hat{\tau}_3 \hat{\sigma}_{0/z} \check{g}\right)\right]. \tag{14}$$

By fitting the experimental $I(V)$ curves with Eqs. (3) and (14) we are able to obtain the different parameters used in the model.

**Model for transverse rectification.** In Fig. 3(c), we calculate the rectification voltage from the experimentally measured $I(V)$ curves at different heating currents $I_H$ using the following theoretical model. The open circuit voltage $V_s$ for the transverse rectifier configuration shown in Fig. 3(a) can be determined by imposing that the total current crossing the tunnel junction is zero:

$$I = \int_0^L i(V(x))dx = 0. \tag{15}$$

Here, the tunnel-current density ($i(x)$) is integrated from 0 to $L$ along the length of the junction. $V(x) = \frac{x}{L} I_H R_x + V_s + V_{inst}$ is the voltage drop between the N and S at $x$ ($0 < x < L$), $R_x$ is the lateral resistance of the junction, and $V_{inst}$ is the instrumental offset, which is obtained from $I(V_{inst}) = 0$ at $I_H = 0$. The first two parts contains two contributions: a larger trivial Ohmic contribution due to the heating current ($\frac{x}{L} I_H R_x$), and a smaller contribution due to the rectification effect ($V_s$). The former is antisymmetric in $I_H$, whereas the latter is symmetric. Therefore, the symmetrized voltage

$$V_{sym} = \frac{1}{2}[V_s(I_H) + V_s(-I_H)] \tag{16}$$

comes from the rectification effect only.

## Data availability

The data that support the findings of this study are available from the corresponding author upon reasonable request.

## Code availability

The code or mathematical algorithm used to generate results that are reported in the paper and central to its main claim are available from the corresponding author upon reasonable request.

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

## Acknowledgements

This work was mainly supported by the EU's Horizon 2020 research and innovation program under Grant Agreement No. 800923 (SUPERTED) funding all the authors. E.S. and F.G. acknowledge the European Research Council under Grant Agreement No. 899315 (TERASEC), and the EU's Horizon 2020 research and innovation program under Grant Agreement No. 964398 (SUPERGATE) for partial financial support. M.S. and E.S. acknowledge partial funding from the European Union's Horizon 2020 research and innovation program under the Marie Skłodowska Curie Action IF Grant No. 101022473 (SuperCONtacts). J.M. acknowledges financial support in the USA by the Army Research Office (grant ARO W911NF-20-2-0061), ONR (grant N00014-20-1-2306), NSF (grant DMR 1700137) and NSF C-Accel Track C Grant No. 2040620. The work of F.S.B., C.R. and M.I. was supported by the Spanish Ministerio de Ciencia e Innovacion (MICINN) through Project PID2020-114252GB-I00 (SPIRIT). F.S.B. acknowledges financial support by the A. v. Humboldt Foundation.

## Author contributions

E.S., N.L., and M.S. performed the experiments and analyzed the data. S. I, M.R, P.V., T.H., and F.S.B. provided theoretical support. J.M. fabricated the Cu/EuS/Al devices and C.G.O, M.I, and C.R. the EuS/Al/AlO$_x$/Co ones. E.S. conceived the experiment together with F.G. who supervised the project. E.S. and M.S. wrote the manuscript with feedback from all authors.

## Competing interests

With the Institute Nanoscienze-CNR, the following authors: E.S., M.S., F.G., P.V., T.T.H., S.I, and F.S.B., have filed a patent (International Application N. PCT/IT2021/000038 "APPARATUS AND METHOD FOR SUPERCONDUCTING DIODE", status: pending, aspect of manuscript covered in patent application: rectification and diode-behavior of the material combinations, architecture and measurement configurations presented). The remaining authors declare no competing interests.
