## [Peer Review File · Nature Communications]

Reviewers' Comments:

Reviewer #2:

Remarks to the Author:

The authors present experimental evidence of direction-selective charge flow in a normal metal/ferromagnetic insulator/superconductor system (Cu/EuS/Al). Substantial rectification occurs at voltage biases less than 1 meV. The underlying physics is related to the spin-dependent particle-hole asymmetry that occurs in the density of states in a superconductor with a spin splitting.

The manuscript is well written and the theoretical modelling is reasonable. I think this manuscript will be of interest in the community, and I would be willing to reconsider a revised version of the manuscript for publication in Nature Communications if the authors can address the following points:

1) In the abstract, "electron-hole asymmetry brake" should be replaced by something like "broken electron-hole symmetry".

2) At the bottom of page 4 and in the caption of Figure 2, there is something inconsistent with the use of I_{Sym} and I_{Antisym} .

- First, it says that "a sizable I_{Sym} already suggests an efficient charge rectification". Shouldn't this be I_{Antisym} ?

- Moreover, the rectification R is defined as $R = I(V) - I(-V)/[I(V) + I(-V)] = I_{\text{Sym}}/I_{\text{Antisym}}$. But this is inconsistent with the definitions given of I_{Sym} and I_{Antisym} . This inconsistency also appears where $R(V)$ is defined in the caption of Figure 2.

- Is this confusion regarding I_{Sym} and I_{Antisym} merely a typo or does it affect any of the results/plots in the paper?

Reviewer #3:

Remarks to the Author:

The manuscript by E. Strambini et al. presents an investigation of the nonreciprocal transport characteristics of a superconducting (SC) tunnel diode, based on a ferromagnetic insulator (EuS) separating a normal (Cu) and a superconducting metal (Al). The rectification is obtained as a result of the spin-splitting of the BCS DOS in Al induced by proximity with EuS, combined to the spin filtering action of EuS itself. The manuscript is timely: in the last months several works have addressed the topic of the current rectification in SC devices. The SC tunnel diode here presented has interesting potential applications for THz signal detection, and the authors quantitatively discuss (Supp. Info.) the maximum resolution and the noise equivalent power (NEP) of a detector based on the mechanism under study.

The experimental results, the model and the supporting information are presented in a clear fashion. The explanation of the reported effects is convincing. Therefore, I support the publication of the manuscript in Nature Communications, provided my questions and comments below are addressed.

1. It is not very clear to me the motivation behind the configuration (ii); more precisely, it is not clear what is the additional information which is provided with respect to the more straightforward experiment in configuration (i). As far as I see, results in (ii) can be explained by the same mechanisms governing (i), but they are just more difficult to interpret owing to the complex current pattern, where charge must enter and exit Al –and this in a distributed way. I think the authors should motivate this point, and highlights the advantages (and disadvantages, if any) of (ii) with respect to (i).

2. This is related to the previous point. In the Methods section (which is part of the main text), in "Models for transverse rectification" there are several quantities that are not defined, adding

confusion in the interpretation of (ii). For the current, there are I and I_H (and it is not clear their relationship with I_{Cu} defined in the main text). V_s is also not very clear, as well as the integration variable s . I think it would be useful either to add a scheme with the model the authors have in mind when performing the integration (with the corresponding labels), or at least to add labels in Fig.3a with some clarification.

3. Are thermoelectric effects also important in configuration (i)? Joule dissipation in Cu should produce heating and thus a temperature difference.

The following points are minor indications.

4. In the literature review section presented in the introduction, I would find appropriate to add references to the rectification effect produced by magnetochiral resistance in superconductors with large spin-orbit. This work was pioneered by the group of Y. Iwasa in Japan. In order to have an up-to-date reference list, I would suggest, e.g., the addition of the following references:
-Wakatsuki et al. Science Advances 3, e1602390 (2017),
-Itahashi et al, Science Advances 6, eaay9120 (2020).

For what concerns *supercurrent rectification*, beside Refs. [7,8,9], I would add the recent following works (preprints) on 2D materials:
-Díez-Mérida et al, arXiv:2110.01067 (2021),
-Bauriedl et al, arXiv:2110.15752 (2021),
-Shin et al, arXiv:2111.05627 (2021).

5.1 I agree very much with the choice of the term superconducting *tunnel* diode in the manuscript title: due to the recent interest in the field of nonreciprocal transport in superconductors, the term "superconducting diode" has been used in different contexts [e.g. (1) magnetochiral anisotropy in the *resistance*, see the works by the group of Iwasa; (2) nonreciprocal *supercurrent*, Ref [8-9] plus the recent preprint works suggested in my previous comment; (3) nonreciprocal Joule dissipation, Ref[7]]. Therefore, I would recommend to use the term *tunnel* in between "superconducting" and "diode" also in the Titles of the Figure Captions (Figs. 1-4).

5.2 Also, I would like to point out that the title "Observation of Superconducting tunnel diode effect" sounds now too similar to "Observation of superconducting diode effect" the title of the seminal paper of Ando et al., Ref [1], Nature 584, 373 (2020). That paper discusses supercurrent rectification, while the present manuscript describes normal current rectification which is boosted by superconductivity plus spin filtering effect. The strong similarity of the two titles might instead suggest that the effects studied in the two works are the same and thus be confusing for the reader.

6. Fig.2b: Most of the information in this graph is contained in the central region. At the same time, most of the graph space is empty. I think it would be then convenient to add an inset (in the upper left or lower right quadrant) with a magnified view of, say, $[(-0.4\text{mV}, 0.4\text{mV}), (-0.03\text{mA}, 0.03\text{mA})]$.

7. Caption Fig. 2. the authors speak about "full numerical calculations". I think it would be useful for the reader to provide the precise section of the Supp. Info. where this can be found.

8. Fig.3a. I'm not sure the electrical scheme is correct. Is the "-" (minus) connection in "V" really connected to the middle of the resistor? From the scheme in Fig1a I would rather say that it is on the left side of it. Also, I think the author have in mind a distributed model of this circuit: if so, this must be clarified.

9. Fig.3c: The graph uses a mixed notation for the unit labels: in the abscissas the μ is next to the numbers (engineering notation), in the ordinates is in the units of the axis title (μV). This must be made uniform, and possibly the choice used for the ordinates should be preferred.

Also, the curves for $+0.1$ and -0.1 T are on top of each other, and it is difficult to understand that there are indeed two curves. It is probably exactly the intention of the authors to show that results for opposite field are similar, but I would try to find a solution that makes clear that there is a curve in the background (perhaps bigger empty symbols for the curve in background?)

10. Fig 4. For panel b, I would use the same choice of color sequence (from cold colors to warm colors when going from low to high T) as in panel a: it makes immediately clear at a glance in which direction the temperature grows.

Response to Referees

We thank all the Referees for reviewing our manuscript and for providing thoughtful comments. Below, we reply point by point to the Referees' comments and suggestions. The resubmission also includes a separate file, where the changes to the manuscript are marked in red for convenience.

REVIEWER COMMENTS

Reviewer #2 (Remarks to the Author):

The authors present experimental evidence of direction-selective charge flow in a normal metal/ferromagnetic insulator/superconductor system (Cu/EuS/Al). Substantial rectification occurs at voltage biases less than 1 meV. The underlying physics is related to the spin-dependent particle-hole asymmetry that occurs in the density of states in a superconductor with a spin splitting.

The manuscript is well written and the theoretical modelling is reasonable. I think this manuscript will be of interest in the community, and I would be willing to reconsider a revised version of the manuscript for publication in Nature Communications if the authors can address the following points:

1) In the abstract, "electron-hole asymmetry brake" should be replaced by something like "broken electron-hole symmetry".

We agree with the reviewer and we rephrase the abstract by substituting "electron-hole symmetry brake" with "broken electron-hole symmetry"

2) At the bottom of page 4 and in the caption of Figure 2, there is something inconsistent with the use of I_{Sym} and I_{Antisym} .

We thank the reviewer for the comment, as discussed below there is a typo in the detailed definition of $R(V) = I(V) - I(-V)/[I(V) + I(-V)]$ in which the sign in the definition is reversed. The correct definition of R is $R(V) = I(V) + I(-V)/[I(V) - I(-V)]$ and is consistent with the use of I_{Sym} and I_{Antisym} . We corrected it in the amended version of the manuscript.

- First, it says that "a sizable I_{Sym} already suggests an efficient charge rectification". Shouldn't this be I_{Antisym} ?

No, I_{Antisym} does not imply rectification. For example, in a linear ohmic resistor in which the $I(V)$ is antisymmetric there is no rectification. This is also consistent with the definition of R that have been used in the text ($R = I_{\text{Sym}}/I_{\text{Antisym}}$)

- Moreover, the rectification R is defined as $R = I(V) - I(-V)/[I(V) + I(-V)] = I_{\text{Sym}}/I_{\text{Antisym}}$. But this is inconsistent with the definitions given of I_{Sym} and I_{Antisym} . This inconsistency also appears where $R(V)$ is defined in the caption of Figure 2.

As mentioned above, the first part of the definition of R was not correct as the sign was reversed. The correct definition is: $R(V) = I(V) + I(-V)/I(V) - I(-V) = I_{Sym}/I_{Antisym}$ and it has been corrected in the new version of the manuscript.

- Is this confusion regarding I_{Sym} and $I_{Antisym}$ merely a typo or does it affect any of the results/plots in the paper?

We apologize for the confusion originating from the typo in the extended definition of $R(V)$ as explained before. The definitions of I_{Sym} and $I_{Antisym}$ are correct as well as the results of the plots that have been calculated considering the right definition of $R(V) = I_{Sym}/I_{Antisym}$.

Therefore, the typo is not affecting any plot of the paper or the conclusions.

Reviewer #3 (Remarks to the Author):

The manuscript by E. Strambini et al. presents an investigation of the nonreciprocal transport characteristics of a superconducting (SC) tunnel diode, based on a ferromagnetic insulator (EuS) separating a normal (Cu) and a superconducting metal (Al). The rectification is obtained as a result of the spin-splitting of the BCS DOS in Al induced by proximity with EuS, combined to the spin filtering action of EuS itself. The manuscript is timely: in the last months several works have addressed the topic of the current rectification in SC devices. The SC tunnel diode here presented has interesting potential applications for THz signal detection, and the authors quantitatively discuss (Supp. Info.) the maximum resolution and the noise equivalent power (NEP) of a detector based on the mechanism under study.

The experimental results, the model and the supporting information are presented in a clear fashion. The explanation of the reported effects is convincing. Therefore, I support the publication of the manuscript in Nature Communications, provided my questions and comments below are addressed.

We agree with the reviewer, supercurrent diodes are an interesting topic that is recently attracting a lot of attention. In the following we address all the constructive considerations raised by the reviewer.

1. It is not very clear to me the motivation behind the configuration (ii); more precisely, it is not clear what is the additional information which is provided with respect to the more straightforward experiment in configuration (i). As far as I see, results in (ii) can be explained by the same mechanisms governing (i), but they are just more difficult to interpret owing to the complex current pattern, where charge must enter and exit Al –and this in a distributed way. I think the authors should motivate this point, and highlight the advantages (and disadvantages, if any) of (ii) with respect to (i).

Configuration (ii) is more complicated than (i) and was mainly introduced to further investigate the capabilities of the device in a detector as it was briefly mentioned in the conclusions: ‘The transverse rectifier benefits from a lower impedance and a direct decoupling between the AC excitation line (the antenna) and the DC sensing line.’

Namely, when used as a detector in the far-infrared regime, the radiation absorption needs to be concentrated near the junction region for optimal quantum efficiency. This can be achieved by preparing separate antennas and matching the impedance of the absorber region with the antenna impedance in the relevant frequency range of interest. Typically, the resistance per area of the junction is rather high so that matching the dissipative part of the impedance would require very large-area junctions. It is then easier to use the configuration (ii) where the impedance matching can be done by mostly disregarding the junction resistance and rather concentrating on the much lower resistance of the metallic wires.

The two configurations are investigated and compared in the supplementary information where we estimate their sensitivity if used in an ideal detector. From the sensitivity we extracted the extrinsic Noise equivalent power of the two configurations coupled to room-temperature voltage or current amplifiers. From this comparison we can conclude that the two configurations have a similar sensitivity at the same boundary conditions (compare for example the orange line in fig.S4e with the gray plot of fig.S5d). The best NEP $\sim 10^{-19} \text{W}/\sqrt{\text{Hz}}$ has been obtained for config (i). Notably, this is just a lower bound as the intrinsic noise in the same temperature range ($\sim 100 \text{ mK}$) would limit the NEP $\sim 10^{-16} \text{W}/\sqrt{\text{Hz}}$ (see section III of the new supplementary information for more details on the estimation).

From this analysis we were also able to quantify thermoelectric effects and we realized the competition between thermoelectricity and transversal rectification in both configurations (config (i) have been included also in the new version of the supplementary information). Relevant when designing detectors, this analysis implies an unwanted region of parameters where the competition leads to a vanishing detector output.

We highlight the advantages and disadvantages between the two configurations in the new version of the manuscript (main text and conclusions, see the changes highlighted in the new version of the manuscript)

2. This is related to the previous point. In the Methods section (which is part of the main text), in “Models for transverse rectification” there are several quantities that are not defined, adding confusion in the interpretation of (ii). For the current, there are I and I_H (and it is not clear their relationship with I_{Cu} defined in the main text). V_s is also not very clear, as well as the integration variable s . I think it would be useful either to add a scheme with the model the authors have in mind when performing the integration (with the corresponding labels), or at least to add labels in Fig.3a with some clarification.

We modified the methods section to be more compliant with the definitions introduced in the main text and in Fig.3a. Following the reviewer's suggestion we introduced more labeling in Fig3a to make the model more clear

3. Are thermoelectric effects also important in configuration (i)? Joule dissipation in Cu should produce heating and thus a temperature difference.

This is a good question. Thermoelectric effects may also affect the output in configuration (i). This happens when we include the voltage drop inside Cu and the charge imbalance close to the junction inside Al, and their effect on heating the junction. The entirely general solution

of the problem would require solving the kinetic equations for position dependent charge and heat imbalances on both sides of the junction. This analysis would include similar parameters as in the case of configuration (ii), but now the resistance of the wires would be directly in series with the tunnel resistance. However, the solution can be simplified when two conditions are satisfied: (a) junction normal-state resistance is higher than the wire resistances, and (b) heat conductivity is dominated by the local electron-phonon and tunneling contributions, which is true for the junction sizes here. In this case we may neglect non-uniformity of current distribution and temperature. Such a thermal model and the balance between thermoelectricity and rectification is discussed in Sec. V of [arXiv:2109.10201].

Using a similar model, we have calculated the IV characteristics of the junction with and without heating effects as shown in the following figure by the blue and orange curves, respectively.

Similarly to config (ii) the net effect of the bias current is an overheating of the S leads, having smaller volume and being decoupled from the phonon bath. This results in a small but not negligible generation of thermoelectric effect, which lowers the net rectification as shown in the following figure where the rectification coefficient R is calculated with and without heating effects.

We thank the reviewer for the stimulating comment and have introduced a discussion on thermoelectric effect in the main text with the extra modeling for config (i) in section IV of the supplementary information.

The following points are minor indications.

4. In the literature review section presented in the introduction, I would find appropriate to add references to the rectification effect produced by magnetochiral resistance in superconductors with large spin-orbit. This work was pioneered by the group of Y. Iwasa in Japan. In order to have an up-to-date reference list, I would suggest, e.g., the addition of the following references:

- Wakatsuki et al. Science Advances 3, e1602390 (2017),
- Itahashi et al, Science Advances 6, eaay9120 (2020).

We thank the reviewer for the suggestion, and we included also these two references to extend the literature review of the introduction

For what concerns *supercurrent rectification*, beside Refs. [7,8,9], I would add the recent following works (preprints) on 2D materials:

- Díez-Mérida et al, arXiv:2110.01067 (2021),
- Bauriedl et al, arXiv:2110.15752 (2021),
- Shin et al, arXiv:2111.05627 (2021).

These new preprints were not present at the submission of the manuscript, and we agree with the reviewer to include them in the new version of the manuscript.

5.1 I agree very much with the choice of the term superconducting *tunnel* diode in the manuscript title: due to the recent interest in the field of nonreciprocal transport in superconductors, the term “superconducting diode” has been used in different contexts [e.g. (1) magnetochiral anisotropy in the *resistance*, see the works by the group of Iwasa; (2) nonreciprocal *supercurrent*, Ref [8-9] plus the recent preprint works suggested in my previous comment; (3) nonreciprocal Joule dissipation, Ref[7]). Therefore, I would recommend to use the term *tunnel* in between “superconducting” and “diode” also in the Titles of the Figure Captions (Figs. 1-4).

We agreed and followed the suggestions of the reviewer.

5.2 Also, I would like to point out that the title "Observation of Superconducting tunnel diode effect" sounds now too similar to "Observation of superconducting diode effect" the title of the seminal paper of Ando et al., Ref [1], Nature 584, 373 (2020). That paper discusses supercurrent rectification, while the present manuscript describes normal current rectification which is boosted by superconductivity plus spin filtering effect. The strong similarity of the two titles might instead suggest that the effects studied in the two works are the same and thus be confusing for the reader.

The terms “tunnel” in the title as recognized by the reviewer in the previous comment should be strong enough to discriminate between the two papers. Still, if the two sound too similar we will change it with the new title: “Superconducting spintronic tunnel diode.”

6. Fig.2b: Most of the information in this graph is contained in the central region. At the same time, most of the graph space is empty. I think it would be then convenient to add an inset (in the upper left or lower right quadrant) with a magnified view of, say, $[(-0.4\text{mV}, 0.4\text{mV}), (-0.03\text{mA}, 0.03\text{mA})]$.

By following the suggestion of the reviewer we modified Fig.2b with the inclusion of an inset.

7. Caption Fig. 2. the authors speak about “full numerical calculations”. I think it would be useful for the reader to provide the precise section of the Supp. Info. where this can be found.

We agree with the reviewer and we linked the proper reference in the caption: “Details of the numerical model can be found in the Methods section, specifically in Eqs.~(3) and (14)”.

8. Fig.3a. I’m not sure the electrical scheme is correct. Is the “-” (minus) connection in “V” really connected to the middle of the resistor? From the scheme in Fig1a I would rather say that it is on the left side of it. Also, I think the author have in mind a distributed model of this circuit: if so, this must be clarified.

As properly recognized by the reviewer the schematic presented in Fig.3b is a simplified sketch to describe the symmetric voltage drop originating across the junction due to the distributed model of the circuit. It is not intended to replace the measure scheme shown in Fig1a. We clarified this point in the caption as suggested. The position of the (minus) connection in “V” along the Cu strip is not relevant for the symmetric voltage, then for simplicity in the sketch it has been placed in the middle of the wire.

9. Fig.3c: The graph uses a mixed notation for the unit labels: in the abscissas the μ is next to the numbers (engineering notation), in the ordinates is in the units of the axis title (μV). This must be made uniform, and possibly the choice used for the ordinates should be preferred.

Also, the curves for $+0.1$ and -0.1 T are on top of each other, and it is difficult to understand that there are indeed two curves. It is probably exactly the intention of the authors to show that results for opposite field are similar, but I would try to find a solution that makes clear that there is a curve in the background (perhaps bigger empty symbols for the curve in background?)

We thank the reviewer for the suggestion and we have modified Fig.3c accordingly. We have taken out the engineering notation and made both axes use the scientific notation for the units. In order to show the difference between the two curves we changed the colors (light green and black instead of red and orange) and the symbol type (star and circle) and added a label to each of them.

10. Fig 4. For panel b, I would use the same choice of color sequence (from cold colors to warm colors when going from low to high T) as in panel a: it makes immediately clear at a glance in which direction the temperature grows.

That is a good point. We have made the relevant changes.

LIST OF CHANGES:

All the changes to the manuscript have been highlighted in the comparative document:
Superconducting spintronic tunnel diode_diff.pdf

Reviewers' Comments:

Reviewer #2:

Remarks to the Author:

The authors have addressed the points in the referee reports, and I now recommend this paper for publication.

Reviewer #3:

Remarks to the Author:

The authors have addressed all the questions. I'm satisfied by the new version of the manuscript and by all the answers except one: I still have one doubt about Eq. (15) in Methods.

The authors have fixed the ambiguity with the labels but the idea behind the integral is still not clear. More specifically, the authors still did not explain what it is the physical quantity integrated, and what I (without subscript) is. I will try to be more specific through some questions: in the integral, is the quantity in round bracket next to I an argument of the function $I(V)$ or is it multiplying I ? What is the physical quantity that is integrated in x , a current, current density, a power? Why is the integral set to zero? The authors have an implicit model in mind, which cannot be easily captured by the reader. A few lines of explanation making explicit what I asked above should be sufficient to eliminate any ambiguity.

I support publication of the manuscript in its present version if this issue is clarified.

Response to Referees

We thank all the Referees for the second reviewing our manuscript. Below, we reply point by point to the last Referees' comments and suggestions. The resubmission also includes a separate file, where the changes to the manuscript are marked in red for convenience.

REVIEWERS' COMMENTS

Reviewer #2 (Remarks to the Author):

The authors have addressed the points in the referee reports, and I now recommend this paper for publication.

We thank the reviewer for her/his positive feedback.

Reviewer #3 (Remarks to the Author):

The authors have addressed all the questions. I'm satisfied by the new version of the manuscript and by all the answers except one: I still have one doubt about Eq. (15) in Methods.

The authors have fixed the ambiguity with the labels but the idea behind the integral is still not clear. More specifically, the authors still did not explain what it is the physical quantity integrated, and what I (without subscript) is. I will try to be more specific through some questions: in the integral, is the quantity in round bracket next to I an argument of the function $I(V)$ or is it multiplying I ? What is the physical quantity that is integrated in x , a current, current density, a power? Why is the integral set to zero? The author have an implicit model in mind, which cannot be easily captured by the reader. A few lines of explanation making explicit what I asked above should be sufficient to eliminate any ambiguity.

We thank the reviewer for the positive feedback to most of the questions. The questions about Eq.(15) have been clarified in the new version of the manuscript with a simplified version of the integral to avoid confusion in the different definitions of the current. The current " I " with no label is, by definition, the current that traverse the tunnel junction. In configuration (i) it uniformly tunnels through the junction to generate the voltage drop used for the tunneling spectroscopy shown in Fig.2. In configuration (ii) I is the small ramification of I_H that tunnels from the Cu to the Al and back form the Al to the Cu. Due to the "open circuit" configuration this path is closed and no current can leak out from the Al lead. This condition is described by the integral of I over the tunnel surface that has been set to zero.

The quantity in the round braked next to I is an argument as properly guessed by the reviewer and the physical quantity integrated I is the current, but the condition is valid also for a current density (" i ") as the two are related by the length ($i = I/L$). The voltage drop between the N and S lead has been explicated in $V(x)$, consider also that the superconductor is at constant potential (i.e. V_s does not depend on x), because inside S current is carried by the supercurrents.

For a lack of clarity, we modified eq.15 introducing the current density and defined the argument in the text. With these changes and the few lines of explanation we believe we solved the ambiguity.

I support publication of the manuscript in its present version if this issue is clarified.

We believed that in the new version of the manuscript this issue is clarified.